# A Simulation of the Effects of Badminton Serve Release Height

**John Rasmussen** [1,*,†] and **Mark de Zee** [2,†]

1 Department of Materials and Production, Aalborg University, 9220 Aalborg Øst, Denmark
2 Department of Health Science and Technology, Aalborg University, 9220 Aalborg Øst, Denmark; mdz@hst.aau.dk
* Correspondence: jr@mp.aau.dk; Tel.: +45-9940-9307
† These authors contributed equally to this work.

**Featured Application: The work adds scientific quantification to the dispute leading up to and after the change of the serve rule in badminton in 2018. The work furthermore identifies serve strategies for doubles that might be an attractive tactical variation of the common approach.**

**Abstract:** In this work, we develop and calibrate a model to represent the trajectory of a badminton shuttlecock and use it to investigate the influence of serve height in view of a new serve rule instated by the Badminton World Federation. The new rule means that all players must launch the shuttlecock below a height of 1.15 m, as opposed to the old rule whereby the required launch height was under the rib cage of the server. The model is based on a forward dynamics model of ballistic trajectory with drag, and it is calibrated with experimental data. The experiments also served to determine the actual influence of the new rule on the shuttlecock launch position. The model is used in a Monte Carlo simulation to determine the statistical influence of the new serve rules on the player's ability to perform good serves; i.e., serves with little opportunity for the receiver to attack. We conclude that, for the female player in question, serving below a height of 1.15 m makes it marginally more difficult to perform excellent serves. We also conclude that there might be alternative launch positions that would be less likely to produce the best serves but could be exploited as a tactical option.

**Keywords:** Monte Carlo simulation; shuttlecock; aerodynamics; trajectory

## 1. Introduction

In badminton, each rally begins with a serve. The server's feet are placed inside the service box, at a minimum distance of 1.98 m to the net, and the shuttlecock must land inside the diagonally opposite service box on the receiver's side after clearing the net height of 1.55 m. The game has evolved such that most serves in singles and, in particular, in doubles are short serves. In doubles, the server's feet are placed at the corner of the service box close to the middle of the court and close to the net, and the target of the serve is often the near corner of the receiver's service box; i.e., the shortest possible flight trajectory. The server will typically perform the serve with the arms extended in front of the body to approach the launch position to the net and further minimize the length of the flight trajectory.

In this situation, the launch height becomes an important factor, with a larger height enabling a flatter serve and minimizing the opportunity of the receiver to attack the serve. Therefore, the classical rules of the serve constrain the position of the racket head at impact with the shuttlecock to be below the lower rib of the player. This means that taller players are able to launch the shuttlecock from a higher position than players of smaller statures.

In 2018, the Badminton World Federation (BWF) decided to experimentally change the rules of the serve and introduced a new limitation formulated as follows: "The whole of the shuttle shall be below 1.15 meters from the surface of the court at the instant of being hit by the server's racket" [1]. To assist in the enforcement of the rule, the BWF also introduced a

new apparatus by which the service judge can verify visually that the maximum launch height is not exceeded.

This rule was put in place for the first time during the All England Open Badminton Championships 2018 and sparked much controversy among players. The frame of reference for the launch position changed from a relative measure with respect to the server's rib cage to an absolute measure with respect to the court. While no official explanation was provided, the rule change likely reflected a desire not to disfavor players with smaller statures. The new rule means that taller players must lower their launch position, while some shorter players can raise theirs, and the point of the controversy is whether a relative or an absolute measure is more fair.

The serving situation, particularly in doubles, is stressful because it is easy to attack the serve, and so the serving team will often find themselves in a defensive situation; thus, the serving height influences the balance of the game.

The purpose of this investigation was to provide quantitative measures for the influence of serve height. More precisely, we aimed to answer the following research questions:

1.  How much lower is the new launch position compared with the old one for a particular world-class female doubles player, for whom the rule change has required an unwelcome adjustment?
2.  Can we quantify the effect of the change in launch position on serve quality?
3.  Are there under-exploited serve strategies that may come into play in light of the new serve rule?

As a by-product of the investigations, we also identified aerodynamic parameters for the numerical simulation of shuttlecock flight trajectory at Reynolds numbers typical for short serves.

Previous works on shuttlecock trajectory simulation are limited but have successfully simulated flight trajectories. Texier et al. [2] noted that the flight of a shuttlecock is influenced by drag and different from a theoretical parabola. The subject of different material choices (plastic versus natural feathers) has attracted some attention, and Cooke [3] developed a simulation based on aerodynamic measurements [4] for the flight of different types of shuttlecocks and found agreement with experimental results. Chan and Rossmann [5] investigated a variety of designs and simulated their trajectories based on aerodynamic measurements.

Chen et al. [6] developed an analytical model of the trajectory based on easily quantifiable measures such as the terminal speed of a freely falling shuttlecock as described by Peastrel et al. [7].

McErlain-Naylor et al. [8] fitted a model based on simplified mechanical principles to measure motion capture data of the shuttlecock trajectories of overhand smashes by international badminton players to investigate the effect of the impact location of the shuttlecock on the racket on the shot outcome for badminton smashes. The observed shuttlecock velocities post-impact were 82.1 m/s on average, which is much higher than for serving.

The aforementioned works document that shuttlecock trajectory simulation is feasible but sensitive to input parameters, among which are shuttlecock designs and the physical conditions such as temperature and humidity. We therefore developed a model and calibrated it with experimental serve data for a particular shuttlecock to seek answers to the research questions.

## 2. Materials and Methods

### 2.1. Experiments

One female badminton player, with a stature of 1.78 m and a BWF doubles top-five ranking, was recruited for the experiment. A badminton court with official dimensions [1] was established in a motion capture lab equipped with an eight-camera Qualisys Oqus system operating at 500 Hz. A tournament-grade feathered shuttlecock of brand Forza was

stained with retro-reflective paint on the cork to allow tracking by the infra-red cameras. The mass of the stained shuttlecock was measured to 6.20 g.

The test subject performed regular doubles serves to different court targets; i.e., predominantly short serves. The serves were first performed according to the old serve rule—i.e., keeping the racket head under the test subject's lower rib—and subsequently according to the new rule—i.e., launching the shuttlecock from below 1.15 m in height. In the latter case, the launch position was verified with an authorized apparatus manufactured and provided by the Badminton World Federation. Shuttlecock cork trajectories were recorded and stored on C3D files by the motion capture system. The recorded trajectories were used to distinguish launch positions and calibrate the aerodynamic parameters. Initial data processing was performed by importing C3D files into the AnyBody Modeling System (AnyBody Technology, Aalborg, Denmark), where the trajectory was interpolated and re-sampled, and the launch positions and launch velocity vectors were computed.

### 2.2. Numerical Model

The equations of motion for the ballistic flight of a shuttlecock in its vertical plane of motion are

$$F_D + m\mathbf{g} - m\mathbf{a} = \mathbf{0}, \tag{1}$$

where $\mathbf{g} = (0, -9.81)$ m/s$^2$ is the gravity acceleration vector, $m$ is the shuttlecock mass and $\mathbf{a}$ is the acceleration vector. The buoyancy force was neglected.

The drag force was computed as

$$F_D = -\frac{1}{2}\rho||\mathbf{v}(t)||^{(p-1)}\mathbf{v}(t)C_D A, \tag{2}$$

where $\rho = 1.16$ kg/m$^3$ is the air density, $\mathbf{v}(t)$ is the velocity vector of the shuttlecock, $C_D$ is the drag coefficient and $A$ is the cross-sectional area of the shuttlecock.

The shuttlecock was assumed to be axisymmetric about its velocity vector and to fly with its axis parallel to the velocity vector, resulting in the absence of lift forces.

The Reynolds number of the flight trajectory is

$$Re = \frac{||\mathbf{v}(t)||\rho D}{\mu}, \tag{3}$$

where $D = 0.065$ m is the diameter of the shuttlecock, and $\mu = 1.86 \times 10^{-5}$ kg/ms is the viscosity of air.

Equation (1) is a second-order ordinary differential equation, which we solved by numerical time integration using a fourth-order Runge–Kutta approach implemented in Python. Given the launch position and launch velocity vector as initial conditions, the time integration ran until the $y$ coordinate of the shuttlecock reached zero, at which point the landing position was recorded.

In the absence of lateral forces on the shuttlecock, the flight trajectory is two-dimensional, and it therefore was simulated in a 2D coordinate system. After simulation, the trajectory was rotated into its spatial direction given by the launch velocity vector.

### 2.3. Model Calibration

Equation (2) represents an empirical approximation to a complex flow phenomenon governed by the Navier–Stokes equations. Drag on an object is caused by a combination of (a) friction between the object and the surrounding fluid related to the velocity gradient in the boundary layer and (b) the pressure difference between the upstream and downstream sides of the object related to turbulence in the fluid. In case (a), the drag force is proportional to the flight velocity—i.e., $p = 1$—and in case (b), it is proportional to the square of the velocity—i.e., $p = 2$. Derivations of these basic dependencies are available in many textbooks; for instance, [9]. The shuttlecock has a complex geometry and is not airtight, so it may harbor regions of flow dominated by either (a) or (b). When calibrating $C_D$ and

*A* simultaneously with all the recorded flight trajectories, the results were unsatisfactory when assuming either (a) or (b). An alternative approach would be to include two terms in the drag equation representing (a) and (b), respectively, and require the identification of two corresponding drag coefficients, say $C_{Da}$ and $C_{Db}$. However, the model also calibrates accurately to a constant, non-integer $p$. This has the advantage of retaining a single value for $C_D$ for identification, albeit with the disadvantage of $C_D$ not being dimensionless but having the unit $[(ms)^{(2-p)}]$.

Directly after impact with the racket, the shuttlecock tumbles and turns in the air before it stabilizes into its flying position with the cork leading. This behavior made the recording of the initial velocity vector from the motion capture data subject to considerable uncertainty. Therefore, initial velocity vectors were also subjects of calibration.

Parameters $\rho$, $C_D$ and $A$ multiply to form a common proportionality constant in Equation (2), making their respective influences indistinguishable. Thus, only $C_D$ was included in the calibration, and it effectively contained adjustments for $\rho$ and $A$ as well.

Calibration was formulated as the following optimization problem:

$$\underset{\mathbf{v_{i0}}, C_D, p}{\text{Minimize}} \quad \sqrt{\sum_{i=1}^{n_e} \int_{x_{ie0}}^{x_{ie1}} (y_{is}(x) - y_{ie}(x))^2 \, \mathrm{d}x}, \tag{4}$$

where $n_e = 8$ is the number of empirical trajectories, $(x_{ie}, y_{ie})$, available to calibrate the model, $x_{ie}$ spans from $x_{ie0}$ to $x_{ie1}$. $(x_{is}, y_{is})$ are the simulated trajectories. The initial velocity vectors corresponding to the trajectories are labelled $\mathbf{v_{i0}}$. With eight initial velocity vectors, each comprising two coordinates, and two unknown physics parameters, $C_D$ and $p$, the calibration problem in Equation (4) comprised a total of 18 variables.

### 2.4. Statistical Simulation

One Monte Carlo simulation was performed for each of the identified mean serve heights using the calibrated aerodynamic properties. In the interest of finding attractive serve strategies that are different from the accepted approach, the random number generator used uniform distributions within rather wide intervals. The launch velocity was picked randomly in the range of 4–20 m/s, the launch angle was in the range of 10–30 degrees, and the launch distance to the net was in a range of 1 m on each side of the service line.

Simulated serves that hit the net or landed out of bounds were discarded. The simulation continued until 5000 good serves were recorded for each condition.

The quality of each serve was subsequently evaluated by means of the area-of-attack, defined as the area of the trajectory that was over the net height on the receiver's side, as illustrated in Figure 1. Serves with an area-of-attack below 2% of the mean were placed in a pool of excellent serves for further assessment.

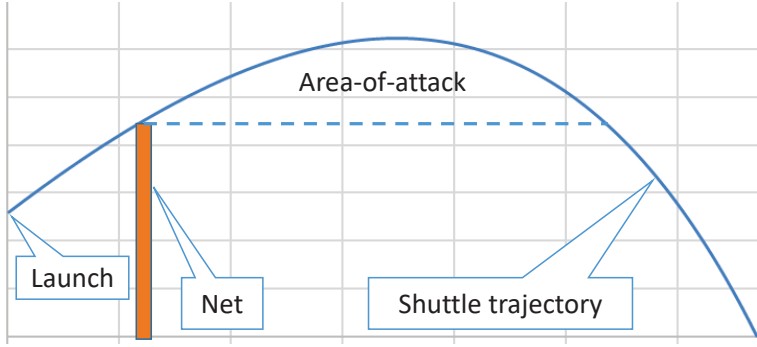

**Figure 1.** Definition of the area-of-attack.

The area-of-attack alone might not be a fully representative metric for the quality of a serve. It can be argued that the time-of-travel of the shuttlecock to the point where the receiver can reach it is also important. Thus, for the selected excellent serves, the

time-of-travel until the shuttlecock was 1 m beyond the net on the receiver's side was computed and correlated against the launch position to reveal possibilities for alternative serve strategies.

## 3. Results

A total of 18 serves were recorded, of which six were faults or inadequately tracked by the motion capture system. Several trials were occluded when the shuttlecock was close to floor level (Figure 2). Eight trajectory recordings were coherent enough for calibration; five recorded launch positions were available for the old serve and nine for the new serve.

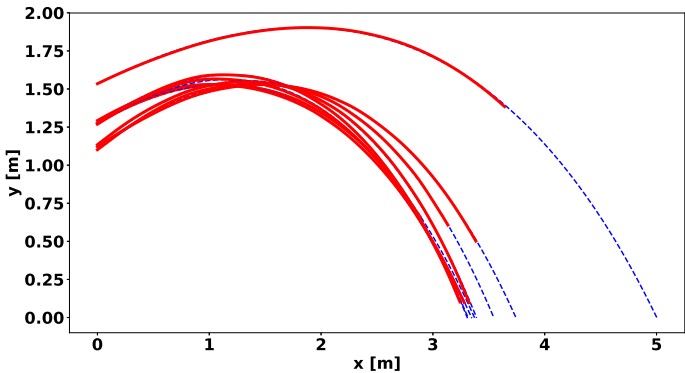

**Figure 2.** Measured (solid) and simulated (dashed) trajectories after calibration. The short trials were launched from the old and new heights, respectively. The long serve was launched without regard for the origin.

### 3.1. Launch Position

The mean of the old launch height of the shuttlecock cork tracked from the motion capture data was 1.18 m, with a range of 1.17 m to 1.20 m.

The mean launch height of the new serve was 1.02 m, with a range of 0.99 m to 1.04 m. All serves performed at the new launch height were below the specified 1.15 m because the rule applies to any part of the shuttlecock and not just to the cork. In the following, we simulate serve conditions for shuttlecock mean heights of 1.18 m and 1.02 m for the old and new serve positions, respectively.

### 3.2. Calibration

The processed motion capture data revealed launch speeds up to 9.5 m/s and trajectory speeds down to 4.5 m/s, creating Reynolds numbers between $1.65 \times 10^4$ and $3.92 \times 10^4$.

The optimization of the difference between measured and simulated trajectories was able to reduce the squared sum of integrals of trajectory differences, as shown in Equation (4), to $7.3 \times 10^{-3}$ m$^2$; i.e., the simulated trajectories were largely indistinguishable from their measured counterparts, as illustrated in Figure 2. The resulting physics parameters were $p = 1.466$ and $C_D = 1.310$ (ms)$^{0.534}$. The stained shuttlecock cork was less visible to the cameras than regular optical markers, so the cameras tended to lose track of the shuttlecock at the lower stages of its flight, while the simulation continued until landing.

### 3.3. Monte Carlo Simulation

The 5000 good serves from 1.18 m and 1.02 m, respectively, are illustrated in Figures 3 and 4.

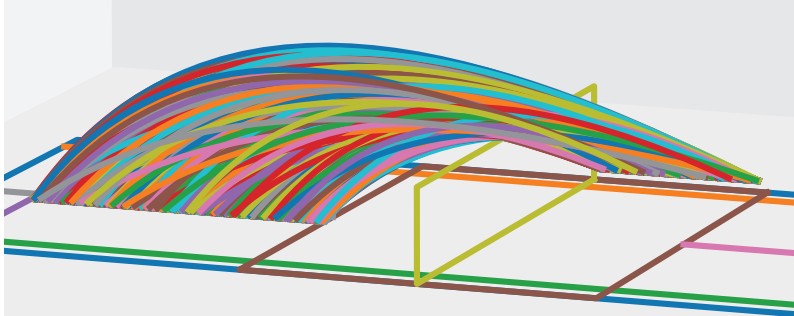

**Figure 3.** Five thousand good serves launched (trajectories from right to left) with randomized launch positions, directions and speeds from the serve height of 1.18 m.

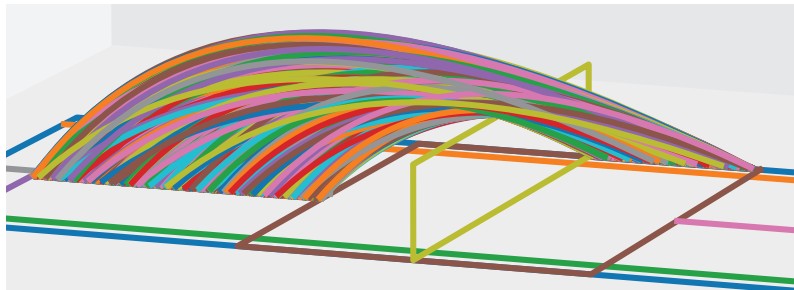

**Figure 4.** Five thousand good serves launched (trajectories from right to left) with randomized launch positions, directions and speeds from the serve height of 1.02 m.

The excellent serves according to the area-of-attack amounted in total to 190 (Figure 5), of which 81 were launched from 1.02 m and 109 from 1.18 m. The four best serves were all launched from 1.18 m. Of the 10 best serves, only three were launched from 1.02 m.

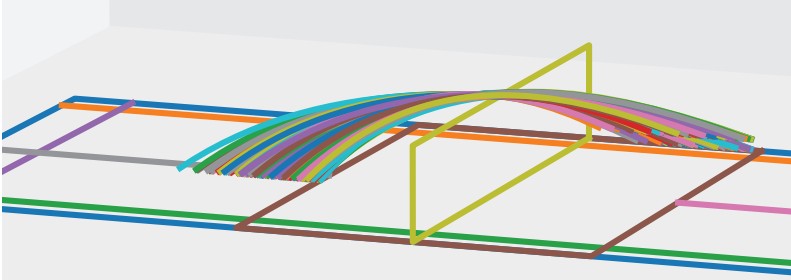

**Figure 5.** The 190 excellent serves better than 2% of the average according to area-of-attack.

Plotting the time-of-flight until 1 m past the net against the launch position for the excellent serves in terms of area-of-attack (Figure 6) revealed a trend of a shorter flight time when launched closer to the net. However, the majority of the 190 excellent serves were launched further back in the court, indicating that the desired flat trajectory is easier to obtain with a higher launch velocity and a flatter angle, and the downside in terms of increased reaction time for the receiver is modest. The results also indicate that the strategy of minimizing the launch position's distance to the net is less beneficial when serving from a lower height.

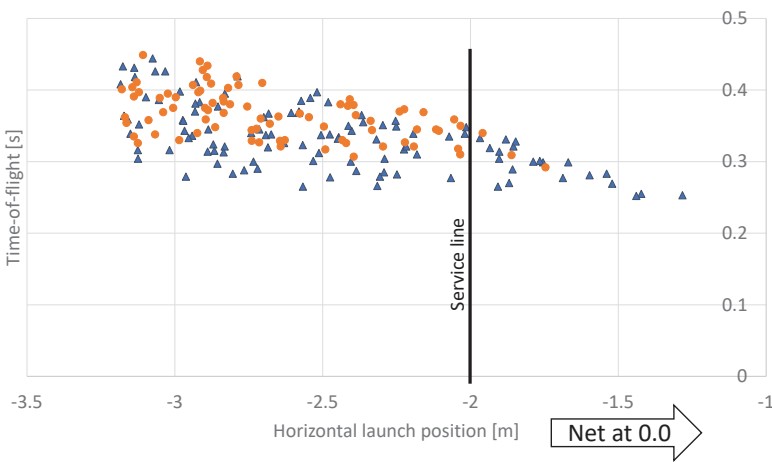

**Figure 6.** The time-of-flight of the excellent serves plotted against launch position. Triangles: old serve height. Circles: new serve height.

## 4. Discussion

The number of successfully recorded trials was lower than originally planned due to unexpected difficulties in the tracking of the stained shuttlecock. A better statistical foundation for establishing the effective launch heights would be desirable.

### 4.1. Calibration

The calibrated trajectories presented in Figure 2 indicate that the model was capable of simulating the flight trajectory with good accuracy within the investigated range of Reynolds numbers. It was remarkable that the drag was proportional to the velocity in a power of $p = 1.466$, when $p = 2$ is often assumed. This indicates that a significant portion of the drag on a badminton shuttlecock at low velocities derives from friction.

Cooke [4], by means of wind tunnel experiments, found a stable $C_D$ around 0.5 for a Reynolds number interval of $[1.3 \times 10^4, 19 \times 10^4]$, albeit with a different definition of the cross-sectional area of the shuttlecock, supporting the assumption of $p = 2$. Chan and Rossmann [5] found drag coefficients around 0.6 and Texier et al. [2] measured $C_D$ up to 0.8.

Apart from different cross-sectional area definitions, the difference in these findings may be due to the low-velocity interval of the present investigation, to a different air density or viscosity, to increased drag caused by the axial rotation of the shuttlecock [10] or to not considering the lift effect and increased drag caused by the deviation of the shuttlecock's centre axis from its velocity vector. Hasegawa et al. [11] observed, after conducting wind tunnel experiments, that "the value of drag coefficient for a shuttlecock without gaps is significantly smaller than that for a standard shuttlecock". They found that the flow of air through the feathers into the shuttlecock increased the drag. In the present investigation, this flow in confined spaces may be a contributor to the friction part of the drag.

The findings cannot be extrapolated to higher-velocity strokes such as clears and smashes without further investigation, but the simulated trajectories indicate that the simulation is valid for the investigated case.

The added paint on the shuttlecock cork increased the mass of the shuttlecock by some fractions of a gram and offset the center-of-mass, which may have led the experimental data to deviate from unstained shuttlecocks.

### 4.2. Monte Carlo Simulation

The visual impression of excellent serves in terms of the area-of-attack (Figure 5) indicates that this criterion is relevant in terms of the nature of the sport. Of these 190 serves, a small majority of 57% were launched from the old position, indicating that the new serve position makes it slightly harder for the tested player to perform excellent serves. This

notion is corroborated by the finding that really excellent serves in terms of time-of-flight were predominantly launched from the old position (Figure 6).

The time-of-flight investigation revealed, surprisingly, that a short time-of-flight is only weakly dependent on the proximity of the launch position to the net, and that there were excellent serves with a short time-of-flight launched from further back in the court. This indicates that there might be under-exploited serve strategies from positions further back in the court.

We can expect that taller players will be more affected by the rule change, and that the new rule may be a significant benefit for very short players.

The validity of the statistical approach for the case at hand can be debated. An alternative method could be the optimization of serve quality as a function of the free parameters, similar to the approach by Sørensen and Rasmussen [12] for association football. If successful, such an approach would determine the best serve for each of the two conditions, and they could subsequently be compared. Realistic optimization, however, would require the consideration of sensitivity to noise in the neuromuscular system, as discussed by McErlain-Naylor et al. [13], and the difficulty of learning a new movement pattern in order to cope with the new serve height. The strategy of the elite player in the present study, when serving from the uncustomary lower height, was to flex her knees to retain the familiar movement pattern in her upper body. The statistical model of the present study revealed that there were many serves of almost identical quality launched from very different positions. In this situation, the statistical opportunity to perform a good serve is more important than finding the absolute optimum.

## 5. Conclusions

Every branch of elite sports is a highly optimized activity. Without doubt, this is also the case for badminton and may be the reason why the new serve rule has been met with much criticism by several top players.

The simulations show that the new rule does indeed disadvantage tall players to some extent. However, it is likely that tactical modifications and the learning of new techniques will gradually soften the consequences for the affected players and perhaps lead to the advent of new serve strategies.

In a wider perspective, the flight of objects is important in many branches of sport. Serves are an integral part of the game in volleyball, squash, table tennis and tennis. In each of these sports, the serve is subject to tactical choices and can decide the match. In tennis, the serve is considered the most important stroke [14], and elite players use combinations of speed, spin and placement to optimize the outcome of their serves. Although the aerodynamics of spinning tennis balls is different from that of badminton shuttlecocks, the tactical considerations concerning the risk of failed serves and the opponent's ability to return the serve are similar. A combined numerical/statistical simulation similar to the approach presented in this paper may be used to elucidate beneficial tactical choices in tennis and possibly in other sports as well.

**Author Contributions:** Conceptualization, J.R. and M.d.Z.; methodology, J.R. and M.d.Z.; software, J.R.; validation, J.R. and M.d.Z.; formal analysis, J.R.; investigation, J.R. and M.d.Z.; data curation, M.d.Z.; writing—original draft preparation, J.R.; writing—review and editing, J.R. and M.d.Z. All authors have read and agreed to the published version of the manuscript.

**Funding:** This research received no external funding.

**Institutional Review Board Statement:** The experiment is allowed under the Ethical Review Board of Northern Jutland, Denmark.

**Informed Consent Statement:** Written informed consent was obtained before the experiment.

**Acknowledgments:** The authors wish to thank Christinna Pedersen and Allan Rossen for their kind assistance.

**Conflicts of Interest:** The authors declare no conflict of interest.

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
