# Peer review of "A Simulation of the Effects of Badminton Serve Release Height"

_applsci, doi:10.3390/app11072903_

Round 1
Reviewer 1 Report
Manuscript has an uncertain hypothesis with a shortage of supporting evidence (Poor Analysis) as well as doesn't add value to the journal
Author Response
We appreciate your time and effort. The manuscript has been modified on several points according to detailed response of the other two reviewers. We hope this will improve your impression of the work.
Reviewer 2 Report
In this paper, authors develop and calibrate a model to represent the trajectory of the badminton shuttle cock and use it to investigate the influence of serve height in view of a new serve rule instated by the Badminton World Federation.
The article is very well written and structured, and it really provides new scientific knowledge to the field of study. It would have an undeniable interest for any reader of Applied Sciences.
However, I have a minor comment that may help to improve the manuscript:
- Conclusion section is very short. Authors could expand it by commenting future lines of research related with this study, or other sports in which a study similar to this could be conducted (tennis?).
Author Response
We thank you sincerely for your time, effort and very constructive input. We have followed your advice and have added perspectives for this approach when used in other fields of sport in general and tennis in particular to the conclusion.
Reviewer 3 Report
General comments:
The manuscript was a numerical simulation of the effect of badminton serve launch height (recent rule change) on serve outcome. The manuscript was well-written and enjoyable to review. The methods were thorough, representing a good addition to the literature and addressing a genuine applied question with clear practical implications. There are a limited number of areas in which I believe the conditions used within simulations could be improved (or justified more) which I will describe below. If the authors wish to make changes then this may require rerunning some simulations but should be relatively straightforward.
Comment 1: I believe the heights used for simulations could be improved (or greater justification provided). If the study compares extreme heights (highest height under old rules vs lowest height under new rules) then it is not investigating the effect of the rule but rather the effect of serve height at two extremes. If the authors wish to compare the effects of the rules then it would seem more appropriate to use either the average experimental height under each rule condition or to sample from the observed distribution of heights under each condition. On this point, what is the justification for using uniform distributions for the random number generator? How does this compare to the observed experimental distributions?
Comment 2: The ‘area-of-attack’ is used as a measure of serve quality. This is well justified in the manuscript. However, the use of the top 1% of serves (drawn from the uniform distributions from two arbitrary extreme serve heights which may not represent average heights under either rule) as criteria for an ‘excellent serve’ could be improved (or justified more). It would be very useful to include an indication of what value is ‘good enough’ or practically meaningful. This would facilitate an investigation of the proportion of serves meeting this threshold. For example, in lines 196-197, are these differences practically meaningful or are they so small that they are practically meaningless? Likewise, if the top 1% are considered ‘excellent’ then are the next 1% different in a practically meaningful way or would they also be considered ‘excellent’?
Specific comments:
Title: The current title seems vague and does not mention the release height, simulations, or that the effects of the serve release height are being investigated. Perhaps something like ‘A simulation of the effects of badminton serve release height’.
‘Shuttlecock’ should be one word throughout and not ‘shuttle cock’ – this is currently inconsistent throughout the manuscript.
Keywords: I believe these should be different to the words used in the title? This particular journal may have different guidelines but it is worth checking.
Introduction:
Line 29: ‘greater height’ is better than ‘larger height’
Lines 36-38: It would be useful to cite the source of the rule here.
Lines 46-50: These two consecutive single sentence paragraphs seem odd and could perhaps be combined or merged with other paragraphs.
Lines 53-54: I am happy with the application to a single elite player, but I believe greater justification for this approach should be included within the manuscript.
Lines 61-71: Greater detail of the methods and parameters used in these previous approaches (and possibly their limitations) would improve the Introduction. Outside of the simulation literature, curve equations based on similar (albeit slightly simplified) mechanical principles have also been fit to experimental shuttlecock displacement data of elite athletes: https://doi.org/10.1080/02640414.2020.1792132 - that particular method was used to address the limitations immediately after impact similarly to your approach in lines 133-140.
Materials and Methods:
Line 79: Is anonymity required? If so, this line may need changing to protect the anonymity of the player.
Results:
Lines 176-177: What were the average serve heights (and standard deviation or interquartile range)? This seems far more informative than the range of extreme values observed.
Line 204: ‘less attractive’ – the study design does not isolate the intentions of the athlete and so ‘less possible’ or ‘less achievable’ may be equally valid suggestions here.
Figure 6: The two colours are difficult to distinguish in the black and white version so this may be worth altering. Also, specifying the actual heights used in the figure heading would be beneficial seeing as the heights used were not the ‘old serve height’ and the ‘new serve height’.
Discussion and Conclusions:
Lines 256-267: As discussed in a recent review paper as part of this same Special Issue (https://doi.org/10.3390/app11041450) the true ‘optimal’ technique for any given athlete (and their ability to learn a new technique) is dependent upon a number of additional factors. The single ‘optimal’ performance cannot be replicated precisely and so the true optimum is likely the technique that maximises success under inherent noise in the neuromuscular system. Likewise, ‘intrinsic dynamics’ are important – i.e. the preferred nodes of coordination or coordination tendencies that exist (often spontaneously or as a result of previous activity) in a movement system at the onset of learning. This is particularly important given the discussion of transitioning from one trained movement pattern (under the old rules) to a new one (under the new rules). The likelihood of the individual adopting and reliably reproducing any theoretical optimum serve solution may be dependent upon both the stability of the attractor corresponding to the individual’s current technique and its proximity within the dynamic attractor landscape to that corresponding to the theoretical optimum technique. I like your suggestion within this section for future directions and possible applications – I am simply suggesting that the current concise wording may make the application seem much more attainable or straightforward than it is in reality.
Round 2
Reviewer 3 Report
The authors have addressed each of my previous comments satisfactorily. I have enjoyed reviewing this paper and thank the authors for their considerable work.